# Snf2 Proteins Are Required to Generate Gamete Pronuclei in *Tetrahymena thermophila*

**DOI:** 10.3390/microorganisms10122426

**Published:** 2022-12-07

**Authors:** Yasuhiro Fukuda, Takahiko Akematsu, Hironori Bando, Kentaro Kato

**Affiliations:** 1Graduate School of Agricultural Science, Tohoku University, Osaki 989-6711, Miyagi, Japan; 2Department of Biosciences, College of Humanities and Sciences, Nihon University, Tokyo 156-8550, Japan

**Keywords:** *Tetrahymena thermophila*, Snf2 family proteins, chromatin remodeling, pronuclei, DNA repair, Iswi, Rad5, euchromatin

## Abstract

During sexual reproduction/conjugation of the ciliate *Tetrahymena thermophila*, the germinal micronucleus undergoes meiosis resulting in four haploid micronuclei (hMICs). All hMICs undergo post-meiotic DNA double-strand break (PM-DSB) formation, cleaving their genome. DNA lesions are subsequently repaired in only one ‘selected’ hMIC, which eventually produces gametic pronuclei. DNA repair in the selected hMIC involves chromatin remodeling by switching from the heterochromatic to the euchromatic state of its genome. Here, we demonstrate that, among the 15 *Tetrahymena* Snf2 family proteins, a core of the ATP-dependent chromatin remodeling complex in *Tetrahymena*, the germline nucleus specific Iswi in *Tetrahymena* IswiG^Tt^ and Rad5^Tt^ is crucial for the generation of gametic pronuclei. In either gene knockout, the selected hMIC which shows euchromatin markers such as lysine-acetylated histone H3 does not appear, but all hMICs in which markers for DNA lesions persist are degraded, indicating that both IswiG^Tt^ and Rad5^Tt^ have important roles in repairing PM-DSB DNA lesions and remodeling chromatin for the euchromatic state in the selected hMIC.

## 1. Introduction

The model unicellular eukaryotic ciliate protozoan *Tetrahymena thermophila* (hereafter referred to as *Tetrahymena*) stably maintains two morphologically and functionally differentiated nuclei within a single cell [1,2]. The large, transcriptionally active macronucleus (MAC) contains the somatic genome, whereas the small, diploid micronucleus (MIC), which is mostly transcriptionally inert, contains the germline genome. The phenotype of a cell depends on the genetic constitution of its MAC, whereas only the MIC genome is transmitted to progeny MACs and MICs during sexual reproduction called conjugation (Appendix A). When conjugation initiates between two cells of different sexes/mating types, their MICs undergo synchronous meiosis. During meiotic prophase, the MICs elongate to form bivalent chromosomes without synaptonemal complex formation [3]. Thereafter, two continuous nuclear divisions take place to form four identical haploid MICs (hMICs) that are in the G2 phase of the cell cycle, which is concomitant with anaphase II [4]. All four hMICs undergo post-meiotic DNA double-strand break (PM-DSB) formation, which is induced by a timely and functionally different manner in the meiotic DSB formation [3]. DNA lesions in hMICs correlate with the appearance of γ-H2AX foci [5], which are markers of DSBs [6]. The γ-H2AX foci disappear only from one hMIC, and this occurs at the same time as the specific localization of the DNA repair proteins; DNAPKcs are involved in DNA repair by non-homologous end-joining and Rad51 involved in recombinational repair to the nucleus [5]. Only this hMIC is selected to form the gamete pronucleus, whereas the three unselected hMICs with persistent γ-H2AX foci are degraded through autophagy mediated by autophagy-related genes (*ATGs*) [7,8]. Attenuated PM-DSB formation by loss of topoisomerase II orthologs culminates in autophagy for all hMICs [5], strongly suggesting that hMIC selection involves self-inflicted DNA damage in all hMICs followed by DNA repair in only one. The selected hMIC undergoes an additional nuclear division, known as gametogenic mitosis, to produce gametic pronuclei [5]. One of the pronuclei migrates to the partner cell to fertilize its stationary pronucleus, whereas the other becomes fertilized by the migratory pronucleus of the partner cell. This reciprocal pronuclear exchange takes place at the conjugation junction where the plasma membranes of conjugating cells are fused [9] and lead to the formation of zygotic nuclei from which both the progeny MACs and MICs differentiate in both partners.

Once selection takes place, the selected hMIC acquires special properties that are not seen in the other three unselected hMICs. For example, the periphery of the selected hMIC is marked by the transmembrane protein Semi1, which mediates the attachment of the nucleus to the conjugation junction [10]. In addition, Semi1 recruits a zinc finger protein Zfr3 to the selected hMIC, which is required for the reciprocal exchange of pronuclei [10]. When it comes to the chromosomes, the disappearance of the γ-H2AX foci from the selected hMIC occurs simultaneously with histone H3 acetylation at lysine 56 (H3K56ac) [5], which is an epigenetic marker of reconstituted chromatin on nascent DNA [11], and the histone H3-H4 chaperone Asf1 specifically localizes to the selected hMIC [5]. Histone H3 also becomes acetylated at lysine 18 (H3K18ac) and 27 (H3K27ac) [5], which are strongly enriched in euchromatin [12,13]. Additionally, a protein containing a high mobility group (HMG) box domain which decreases the compactness of the chromatin fiber [14,15] is abundantly expressed in the nucleus [16]. These findings suggest that the repair of PM-DSBs allows changes in the nucleosome composition to produce fertile gametes. A recent study shows that Nrp1, a NASP-related protein in *Tetrahymena*, interacts with Asf1 and core histones [17,18]. Knockdown of *NRP1* leads to post-meiotic arrest where the γ-H2AX foci persist in all four hMICs [17]. Canonical histones around PM-DSB lesions may be removed either alone or together with chromatin remodelers or histone chaperones [19] to allow access by DNA repair machinery.

The Snf2 family proteins are core to forming chromatin remodelers which have the capacity to add, slide, or eject nucleosomes. The Snf2 family proteins produce energy by hydrolysis of ATP and the remodelers utilize the energy for remodeling activities [20]. The Snf2 family proteins are classified into six groups comprising 24 subfamilies that correlate with functional characteristics and evolutionary phylogenetic relationships [21,22]. The *Tetrahymena* genome possesses multiple genes encoding the Snf2 family proteins [23,24]. Among them, Brg1^Tt^, Chd3^Tt^, and Chd7^Tt^ have been investigated [23,24,25]. These three Snf2 family proteins are abundantly and exclusively expressed in the MAC in vegetative cells. Their genes are essential for cell viability, and these proteins can interact with modified histone proteins directly and indirectly [23,24,25]. They have therefore been regarded as participating in transcription regulation for cell maintenance. Brg1^Tt^, Chd3^Tt^, and Chd7^Tt^ also appeared in the progeny MAC Anlagen during conjugation and have been suggested to play roles in the genome rearrangement for progeny MAC formation [23,24]. However, it is yet to be indicated which Snf2 family protein appears in the selected hMIC, and it is yet to be elucidated whether these Snf2 family proteins are involved with the hMIC selection involving epigenetic modifications.

Here, we focused on *Tetrahymena* Snf2 family proteins that are expressed during the early to mid-phase of conjugation and whose subcellular localization has not previously been elucidated. Our investigation revealed that four Snf2 family proteins (one of two Iswi; IswiG^Tt^, Rad54^Tt^, Rad5^Tt^, and Lodestar^Tt^) appeared in the selected hMIC and that both IswiG^Tt^ and Rad5^Tt^ were required for the hMIC selection followed by gametic pronuclei generation. In the *ISWIG^Tt^* and *RAD5^Tt^* knockout cells, γ-H2AX foci persisted in all hMICs, and none of the epigenetic markers for euchromatin formation appeared in any hMICs. These findings provide keys to understanding the mechanism of hMIC selection in *Tetrahymena*.

## 2. Materials and Methods

### 2.1. Culture Methods and the Initiation of Mating

Two *Tetrahymena* wild type strains CU427 (*chx1-1*/*chx1-1* (*CHX1*; cy-s, VI)) and CU428 (mpr1-1/mpr1-1 (MPR1; mp-s, VII)) were obtained from the *Tetrahymena* Stock Center, Cornell University, Ithaca N.Y. (http://tetrahymena.vet.cornell.edu/). Cells were grown at 30 °C in super proteose peptone (SPP) medium [26] containing 2% protease peptone (Becton Dickinson and Company, Tokyo, Japan), 0.1% yeast extract (Becton Dickinson and Company), 0.2% glucose (Nacalai Tesque INC., Kyoto, Japan) and 0.003% Fe-EDTA (DOJINDO Laboratories, Kumamoto, Japan) with gentle shaking. To make cells competent for conjugation, cells at the mid-log phase were washed with 10 mM Tris-HCl pH 7.6, resuspended in 10 mM Tris-HCl pH 7.6, and starved at 30 °C. To initiate conjugation, equal numbers of cells of two different mating types were mixed and incubated at 30 °C.

### 2.2. Database Searching and Motif Scanning

The Snf2 family proteins are characterized by harboring both the SNF2_N domain (PF00176) and HELICc domain (SM00490) [21,22]. To find *Tetrahymena* orthologs to Snf2 family proteins, we used amino-acid sequences of the domains from representative mammalian and yeast Snf2 family proteins as query sequences in the BLASTp search in the *Tetrahymena* Genome Database (TGD; http://ciliate.org). We retrieved 15 proteins. The NCBI Conserved Domain Search (https://www.ncbi.nlm.nih.gov/Structure/cdd/wrpsb.cgi) and the MOTIF Search (https://www.genome.jp/tools/motif/) were utilized to predict conventional domains/motifs.

### 2.3. Phylogenetic Analysis

The collected sequences were aligned using the multiple sequence alignment program MAFFT v7.294b [27] with the global pair and maxiterate options. The aligned amino-acid sequences corresponding to conserved SNF2_N and HELICc domains were cropped, and all gaps were eliminated. The remaining 329 sites with 126 taxa were utilized for maximum likelihood (ML) phylogenetic tree reconstruction. For ML calculation, the best substitution model and optional parameters were evaluated using Aminosan [28], and LG+FC+I+G was suggested as the best setting. The maximum likelihood ML phylogenetic relationships were calculated using RAxML-NG v. 1.1 [29], and 100 replicated trees were reconstructed from the same model to evaluate the bootstrapping value.

### 2.4. Construction of C-Terminal EGFP-Tagging and N-Terminal mCherry-Tagging Vectors

For the expression of IswiS^Tt^-EGFP, IswiG^Tt^-EGFP, Mot1^Tt^-EGFP, Rad5^Tt^-EGFP, Rad16A^Tt^-EGFP, Rad54^Tt^-EGFP, Lodestar^Tt^-EGFP, and pSmarcal1^Tt^-EGFP, C-terminal tagging of endogenous proteins was conducted using a knock-in strategy (Appendix A) [30]. Approximately 2 kb from the 3′ end of the coding sequence and 2 kb from a downstream region were amplified from genomic DNA of wild-type CU427 cells with PrimeSTAR Max DNA Polymerase (TaKaRa Bio Inc., Shiga, Japan) and the primer sets listed in Appendix A. Amplified fragments were cloned into the pEGFP-NEO4 (GenBank: AB570109.1) plasmid using the NEBuilder HiFi DNA Assembly Cloning Kit (New England Biolabs Japan Inc., Tokyo, Japan).

The procedure for DNA transfection into the MAC in starved cells is performed in a similar manner to the DNA transfection into conjugating cells [31,32]. The details of the procedure are described in Appendix A. It should be noted that, in the DNA transfection targeting the MAC locus with electroporation, the transformation efficiency likely increases if the length of the flanking regions for homologous recombination is longer than used with the gene gun (Dr. Masaaki Iwamoto, personal communication). In addition, transformation efficiency seems to vary considerably depending on the targeted loci.

The C-terminal EGFP tagging modules for *Tetrahymena* were amplified from the constructed plasmids (pISWIS^Tt^-EGFP-NEO4, pISWIG^Tt^-EGFP-NEO4, pMOT^Tt^-EGFP-NEO4, pRAD5^Tt^-EGFP-NEO4, pRAD16A^Tt^-EGFP-NEO4, pRAD54^Tt^-EGFP-NEO4, pLODESTAR^Tt^-EGFP-NEO4, and pSMARCAL1A^Tt^-EGFP-NEO4) with PrimeSTAR Max DNA Polymerase and M13 primers. The resulting PCR products were purified with the Monarch PCR and DNA Cleanup Kit (New England Biolabs Japan Inc., Tokyo, Japan). For electroporation, starved *Tetrahymena* cells were washed and resuspended in 10 mM Hepes-NaOH pH 7.5 at 2 × 10^7^ cells/mL. A mixture of approximately 300 μL of starved-cell suspension containing 25–50 µg DNA was transferred into a cuvette with a 0.2 cm gap and pulsed using a Gene Pulser Xcell (Bio-Rad Laboratories, Inc., Tokyo, Japan) with 25 µF, 230 V, 400 Ω at room temperature. The resulting suspension was resuspended in SPP medium for recovery at 30 °C for 3 h, followed by incubation with 1 μg/mL CdCl_2_ for 1 h to activate the *NEO4* cassette. After adding 100 μg/mL paromomycin sulfate (Tokyo Chemical Industry Co., Ltd., Tokyo, Japan), for selection, 200 μL each of the cell suspension was transferred to 96-well plates and incubated at 30 °C. Resistant cells appeared within 3 days and grew in SPP medium containing an increasing concentration of paromomycin sulfate (from 100 to 10,000 µg/mL) to allow for phenotypic assortment [33].

Initially, C-terminal tagging of endogenous proteins was carried out for all Snf2 family proteins that were examined for subcellular localization analysis. Since C-terminal EGFP-tagged proteins were not expressed in two genes, *SWR1^Tt^* and *INO80^Tt^*, we used N-terminal mCherry tagging modules for the genes. For the expression of mCherry-Ino80^Tt^ and mCherry-Swr1^Tt^, N-terminal tagging of exogenous proteins was conducted as demonstrated previously for the expression of EGFP-DNAPKcs (Appendix A) [5]. Approximately 2 kb from an upstream region and the 5′ end of the coding sequences were amplified from genomic DNA of wild-type CU427 cells with PrimeSTAR MAX DNA polymerase and the primer sets listed in Appendix A. Amplified forward and reverse target fragments were cloned into the *Sac*I–*Sal*I and *Spe*I–*Kpn*I sites, respectively, of the backbone plasmid pmCherry-PAC [10] using the NEBuilder HiFi DNA Assembly Cloning Kit. We replaced the PAC-based drug resistance markers which confer cells resistance to puromycin in the plasmids with a paromomycin resistance marker (*NEO5*), excised from pBNMB1-EGFP (a gift from Kazufumi Mochizuki, Institute of Human Genetics, Montpellier, France) with *Sal*I plus *Xma*I (New England Biolabs Japan Inc., Tokyo, Japan), using T4 DNA ligase (New England Biolabs Japan Inc., Tokyo, Japan). For DNA transfection, the N-terminal mCherry tagging modules for *Tetrahymena* were amplified from the constructed plasmids (pmCherry-INO80^Tt^-NEO5 and pmCherry-SWR1^Tt^-NEO5) with PrimeSTAR Max DNA Polymerase and M13 primers. The resulting PCR products were purified and used for electroporation as described above. Resistant cells appeared within 3 days without adding CdCl_2_ and were grown in SPP medium containing an increasing concentration of paromomycin sulfate from 100 to 10,000 µg/mL to allow phenotypic assortment [33]. Protein expression was induced in cells by adding 0.075 µg/mL CdCl_2_ to starved cells.

### 2.5. Fluorescence Microscopy of Living Cells

Sixty minutes before observation, Hoechst33342 solution (H342, Cellstain^®^, DOJINDO Laboratories, Kumamoto, Japan) was added to conjugating cells to reach 50 ng/mL. The cells were concentrated by centrifugation and resuspended in 3% polyethylene oxide (Sigma-Aldrich Japan K.K., Tokyo, Japan) to increase the viscosity of the medium. Seven microliters of the cell suspension were placed onto a glass slide, and a 22 mm square coverslip was gently placed on the slide. Prepared slides were immediately applied for microscopic observation (Olympus BX50 equipped with UPLFLN60X Objective lens: NA = 0.9 and DP71 Digital Camera; Olympus, Tokyo, Japan).

### 2.6. Indirect Immunofluorescence

Cells were fixed with methanol at −20 °C for 1 h. After removal of methanol by centrifugation (3000× *g*, 1 min), the cell pellet was postfixed in 1% formaldehyde (Nacalai Tesque Inc., Kyoto, Japan) in PBS pH 7.5 at 4 °C for 1 h. After removal of formaldehyde by centrifugation (3000× *g*, 1 min), the pellet was washed with PBS three times and incubated for 1 h at room temperature or overnight at 4 °C with primary antibodies. After washing with PBS, cells were incubated with secondary antibodies for 1 h at room temperature in the dark. After washing with PBS, cells were resuspended in PBS containing 1 µg/mL DAPI (Dojindo Laboratories, Kumamoto, Japan), dropped onto a slide, and mounted under a coverslip. Details for antibodies used are presented in Appendix A.

### 2.7. Gene Knockout Constructions for the Snf2 Family Proteins Localizing to the Selected hMIC

Approximately 2 kb sequences upstream (5′) and downstream (3′) of the macronuclear *LODESTAR^Tt^* (TTHERM_00313280), *RAD5^Tt^* (TTHERM_00037210), and *RAD54^Tt^* (TTHERM_00237490) loci were amplified from CU427 genomic DNA with PrimeSTAR Max DNA Polymerase and the primer sets listed in Appendix A. The amplified PCR products were cloned into the pNEO4 (GenBank: EU606202.1) plasmid [34] using the NEBuilder HiFi DNA Assembly Cloning Kit. For DNA transfection, the modules that contain the *NEO4* cassette and the 5′ and 3′ portions of the *LODESTAR^Tt^*, *RAD5^Tt^*, or *RAD54^Tt^* genomic locus for homologous recombination were amplified from the constructed plasmids (pKoLODESTAR^Tt^-NEO4, pKoRAD5^Tt^-NEO4, and pKoRAD54^Tt^-NEO4) with PrimeSTAR Max DNA Polymerase and M13 primers. The resulting PCR products were purified and used for electroporation as previously described. Resistant cells appeared within 3 days and grew in the SPP medium containing an increasing concentration from 100 to 100,000 µg/mL of paromomycin sulfate to allow phenotypic assortment [33]. Replacement of the target loci from the MAC was confirmed by PCR using the primer sets listed in Appendix A.

*ISWIG^Tt^*Δ strains were previously established and are publicly available as *SNF2*Δ strains (SD02188: snf2[∆::neo3/]snf2[∆::neo3]; II and SD02189: snf2[∆::neo3/]snf2[∆::neo3]; VII) in the *Tetrahymena* Stock Center, Cornell University, Ithaca N.Y. (http://tetrahymena.vet.cornell.edu/). We obtained and used both strains in this study. Replacement of *ISWIG^Tt^* loci from the MAC was confirmed by sequencing and PCR using the primer set listed in Appendix A.

### 2.8. Acetic Orcein Stain

Ten microliter cell suspension was pipetted onto a glass slide and air dried. The glass slide was fixed in 3:1 methanol: acetic acid for 5 min, incubated in 5 N HCl for 5 min to degrade RNA, and then rinsed in distilled water for 10 sec. Acetic orcein solution (Sigma-Aldrich Japan K.K.) was applied to the sample, and stained nuclei were observed under light microscopy.

## 3. Results

### 3.1. Tetrahymena Snf2 Family Proteins

Fillingham et al. indicated that *Tetrahymena* has 16 genes encoding Snf2 family proteins [24]. Since that study, the genome annotation of the *Tetrahymena* Genome Database (TGD; www.ciliate.org) has been updated several times [35,36,37,38,39,40]. Using the most recently updated genomic data, we searched for the *Tetrahymena* genes encoding Snf2 family proteins. Snf2 family proteins are characterized by the SNF_N domain in the N-terminal region followed by a HELICc domain in the C-terminal region [21,22]. We explored the TGD for genes encoding proteins containing these two consecutive domains and found 15 genes (Appendix A).

The Snf2 family proteins are classified into six groups comprising 24 subfamilies that correlate with functional characteristics and evolutionary phylogenetic relationships [21,22]. THERM_01245640, TTHERM_00193800, and TTHERM_0049310 were previously identified as *BRG1^Tt^*, *CHD3^Tt^*, and *CHD7^Tt^*, respectively [23,24]. However, for the Snf2 family proteins encoded by other genes, their groups and subfamilies had not been identified. To identify these, we conducted a molecular phylogenetic analysis using the conserved amino acid sequences corresponding to the SNF_N and HELIC_C domains of the *Tetrahymena* Snf2 family protein genes together with various Snf2 family proteins in evolutionarily diverse organisms. The reconstructed ML phylogenetic tree is shown in Figure 1A. Except for the Rad5/16 group, the subfamilies within each group appeared as robustly supported monophyletic groups (Appendix A). Five Snf2 family proteins, including Brg1^Tt^, Chd3^T^t, and Chd7^Tt^, were classified into the Snf2-like group. Brg1^Tt^, Chd3^Tt^, and Chd7^Tt^ fell into the Snf2 subfamily clade, Mi-2 subfamily clade, and Chd7 subfamily clade, respectively. These results were consistent with the previous classification based on their domain structures, strongly indicating that the reconstructed ML tree was valid. The remaining two genes in the Snf2-like group, TTHERM_00137610 and TTHERM_00388250, were identified as the genes encoding Iswi subfamily proteins. Because the proteins encoded by THERM_00137610 and TTHERM_00585520 exclusively localized to the MAC and the MIC, respectively, we refer to the Iswi^Tt^ encoded by THERM_00137610 as IswiS^Tt^; Somatic MAC Iswi^Tt^, and the Iswi^Tt^ encoded by TTHERM_00585520 as IswiG^Tt^; Germline MIC Iswi^Tt^ (Figure 2 and Appendix A). Proteins encoded by TTHERM_00343570 and TTHERM_01546860 belonged to the Swr1-like group. The former was identified as the gene encoding the Ino80 subfamily protein, and the latter was identified as the gene encoding the Swr1 subfamily protein. TTHERM_00237490 was the only gene encoding the Rad54 subfamily. The Snf2 family protein encoded by TTHERM_00313280 was identified as a member of the Mot1 subfamily in the SSO1653-like group. Proteins encoded by TTHERM_00313280, TTHERM_00420480, TTHERM_0037210, and TTHERM_00933250 fell into the Rad5/16-like group. The Rad5/16-like group comprises four subfamilies; the Rad5/16 subfamily, the Ris1 subfamily, the Lodestar subfamily, and the Shprh subfamily. Except for the last subfamily, no supported monophyletic groups representing subfamilies were reconstructed in the ML phylogenetic tree (Appendix A). We could therefore not identify the subfamily of four proteins assigned to the Rad5/16-like group from the phylogenetic analysis. The remaining two genes, TTHERM_01080500 and TTHERM_00627150, were identified as the genes encoding the Smarcal1 subfamily proteins within the distant group. In this study, we refer to the Smarcal1^Tt^ encoded by TTHERM_01080500 as Smarcal1A^Tt^, and the Smarcal1^Tt^ encoded by TTHERM_00627150 as Smarcal1B^Tt^.

The Snf2 family proteins have subfamily-specific accessory domains/motifs, and these accessory domains/motifs are responsible for the functional characteristics of each subfamily [21,22]. We applied two different motif scan tools to predict the accessory domains/motifs of the 15 Snf2 family proteins found in *Tetrahymena*. Figure 1B shows the structure of the predicted domains/motifs. The accessory domains/motifs detected in Brg1^Tt^, Chd3^Tt^, and Chd7^Tt^ were consistent with the structures identified in previous studies. For two Iswi^Tt^, Ino80^Tt^, Swr1^Tt^, and Rad54^Tt^, accessory domains/motifs unique to each subfamily were identified (e.g., Iswi subfamily proteins possessed SANT and SLIDE domains [41]. Ino80 harbored the DBINO domain at the N-terminal end of the SNF2_N domain [42]). These results were consistent with the classification of these Snf2 family proteins based on the molecular phylogenetic analysis. The four Snf2 family proteins belonging to the Rad5/16-like group were identified as subfamilies based on the detected accessory domains/motifs. Proteins encoded in TTHERM_00420480 and TTHERM_00933250 were classified as Rad16^Tt^ in the Rad5/16 subfamily because both proteins harbor a ring finger motif between the SNF2_N domain and the HELICc domain [43,44]. In this study, we refer to the Rad16^Tt^ encoded by TTHERM_00420480 as Rad16A^Tt^, and the Rad16^Tt^ encoded by TTHERM_00933250 as Rad16B^Tt^. TTHERM_00037210 encodes Rad5^Tt^ in the Rad5/16 subfamily because the HIRAN motif was found in the N-terminal region in addition to the Ring finger motif [45]. The remaining gene TTHERM_00313280 was identified as the gene encoding the Lodestar subfamily protein because the protein lacked both the Ring finger and HIRAN motifs.

### 3.2. Snf2 Family Proteins Appearing in the Selected hMIC

To identify Snf2 family proteins that are involved in hMIC selection, the EGFP- or mCherry-tagged constructs were introduced into WT cells to analyze their subcellular localization. Each fusion protein with EGFP or mCherry was expressed from its endogenous MAC locus or from the cadmium inducible *MTT1* promoter [46]. Based on DNA repair and euchromatin formation, the indexes for hMIC selection exclusively appear in one meiotic product that gives rise to gametic pronuclei [5,16,17], we explored Snf2 family proteins that localize to the selected hMIC but not to the other three unselected hMICs. Here, we excluded Brg1^Tt^, Chd3^Tt^, and Chd7^Tt^, because previous studies elucidated that these proteins exclusively appear in the MAC and the progeny MAC Anlagen [23,24]. We also excluded Rad16B^Tt^ and Smarcal1B^Tt^ from subcellular localization analysis for the following reasons. The expression profiles in the TetraFGD (http://tfgd.ihb.ac.cn/) indicate that *RAD16B^Tt^* transcription is weak during both the vegetative phase and conjugation. *SMARCAL1B^Tt^* is exclusively and actively transcribed after 8 h from the initiation of conjugation. Since the selected hMIC appears about 6 h after the initiation of conjugation, we postulated that Smarcal1B^Tt^ does not participate in the appearance of the selected hMIC.

Figure 2 summarizes the result for Snf2 family proteins involved in hMIC selection, and detailed data are presented in Appendix A. Briefly, 10 Snf2 family proteins showed three patterns of nuclear localization when the hMIC selection took place: (1) they appeared in either the MAC (IswiS^Tt^, Ino80^Tt^, Swr1^Tt^, Mot1^Tt^, and Rad16A^Tt^; Appendix A); (2) they appeared in the selected hMIC (IswiG^Tt^, Lodestar^Tt^, and Rad5^Tt^; Appendix A); or (3) they appeared in both nuclei (Rad54^Tt^; Appendix A). Smarcal1A^Tt^ did not localize to any nuclei at the post-meiotic stage but appeared in the MIC undergoing meiotic prophase and in the zygotic nuclei during the 2nd post-zygotic nuclear division (Appendix A). These results suggest that *Tetrahymena* may have acquired multiple Snf2 family proteins that are specialized to manage its nuclear dualism and indicate the possible involvement of four Snf2 family proteins (IswiG^Tt^, Lodestar^Tt^, Rad5^Tt^, and Rad54^Tt^) in the hMIC selection.

### 3.3. Loss of ISWIG^Tt^ and RAD5^Tt^ Causes a Defect in hMIC Selection

To elucidate whether any or all of IswiG^Tt^, Lodestar^Tt^, Rad5^Tt^, and Rad54^Tt^ are involved in the hMIC selection, the phenotype was observed in conjugating cells in which the respective somatic genes were knocked out. In this observation, we used acetic orcein staining, which allows visualizing *Tetrahymena* nuclei with light microscopy [47]. For *ISWIG^Tt^*, somatic knockout cells of mating types II and VII were obtained from the stock center (https://tetrahymena.vet.cornell.edu/). For *LODESTAR^Tt^*, *RAD5^Tt^*, and *RAD54^Tt^*, we created somatic gene knockout mutants for cells of mating types VI and VII. Gene replacement with antibiotic selection with an exogenous *NEO* cassette [34] can lead to completing the replacement of all 50 target gene copies in the polyploid MAC if the target genes are not essential for vegetative growth, whereas incomplete replacement occurs if they are essential. Phenotypic assortment [33] is attributed to the random distribution of allelic copies in a compound MAC by an amitotic nuclear division. PCR analysis following phenotypic assortment (Appendix A) indicated that fragments corresponding to the wild-type loci of *ISWIG^Tt^*, *LODESTAR^Tt^*, and *RAD5^Tt^* were almost replaced with a fragment for the exogenous *NEO* cassette in both of the mating types. This result indicates that these three genes are dispensable in vegetative growth for both mating types. In the case of the *RAD54^Tt^*, however, the PCR analysis following phenotypic assortment showed incomplete replacement of the endogenous loci by the *NEO4* cassette (data not shown), indicating its indispensable role in vegetative cells. We therefore excluded *RAD54^Tt^* from further analysis. To investigate the impact of depletion of IswiG^Tt^, Lodestar^Tt^, and Rad5^Tt^ for the hMIC selection, we fixed pairing mutant cells every hour after the initiation of conjugation until 16 h and stained fixed cells with acetic orcein to observe nuclear behaviors. The representative images are shown in Figure 3A–D. In *LODESTAR^Tt^*Δ pairing cells, hMIC selection took place at the post-meiosis stage, and the selected hMIC underwent gametogenic mitosis followed by karyogamy and progeny nuclear development (Figure 3B). At 10 h after the initiation of conjugation, the percentage of cells that reached the progeny nuclear development stage did not differ between the wild-type and *LODESTAR^Tt^*Δ pairs (Figure 3E), and the aberrant conjugation phenotype was not found in the *LODESTAR^Tt^*Δ pairs (Figure 3A,B,E). In the *ISWIG^Tt^*Δ pairs and *RAD5^Tt^*Δ pairs, their knockout did not affect meiosis: four normal hMICs were generated, as in the wild-type pairs (Figure 3A,C,D). However, none of the hMIC underwent gametogenic mitosis in both *ISWIG^Tt^*Δ and *RAD5^Tt^*Δ pairing cells at the post-meiosis stage: all hMICs migrated to the posterior region of the cytoplasm, similar to unselected hMIC in the wild-type pair (Figure 3C,D). These hMICs had disappeared by 10 h after the initiation of conjugation, likely via autophagy (Figure 3C,D) [7,8], rendering about 43% of *ISWIG^Tt^*Δ pairs amicronucleate and 66% of the *RAD5^Tt^*Δ pairs amicronucleate (Figure 3E). These results suggest that IswiG^Tt^ and Rad5^Tt^ are involved in the hMIC selection.

### 3.4. Depletion of IswiG^Tt^ or Rad5^Tt^ Does Not Affect PM-DSB Formation but Does Affect DNA Repair and Euchromatin Formation Which Concomitantly Occur in the Selected hMIC

We previously demonstrated that attenuated PM-DSB formation causes a failure in the hMIC selection, resulting in the amicronucleate phenotype [5]. To assess whether PM-DSBs were induced in hMICs after meiosis in *ISWIG^Tt^*Δ pairing cells and *RAD5^Tt^*Δ pairing cells, immunostaining of γ-H2AX was performed. γ-H2AX foci appeared in four hMICs in both *ISWIG^Tt^*Δ and *RAD5^Tt^*Δ pairing cells, as wild-type pairing cells (Figure 4A). This result indicates that PM-DSB formation is independent of either IswiG^Tt^ or Rad5^Tt^ expression.

After the hMIC selection, H3K56 acetylation occurs concomitantly with the disappearance of γ-H2AX foci in the selected hMIC [5]. In addition, H3K18ac, an epigenetic marker representing an enrichment of the euchromatin state appears in the nucleus [5]. We also found that H3K9ac, another marker of euchromatin formation [48], appears in the selected hMIC (Figure 5). These results indicate that DNA repair involving switching from a heterochromatic to euchromatic chromatin structure occurs in the selected hMIC [5]. At 8 h after the initiation of conjugation, four hMICs in both *ISWIG^Tt^*Δ pairing cells and *RAD5^Tt^*Δ pairing cells were undergoing degradation (Figure 3C,D). γ-H2AX foci persisted in their degrading hMICs (Figure 4B), suggesting that none of the hMICs in both mutants underwent DNA repair. As additional evidence for this, we performed immunostaining for H3K9ac, H3K18ac, and H3K56ac. As expected, histone H3 acetylation at K9, K18, and K56 was not detected from four hMICs which were undergoing degradation in both *ISWIG^Tt^*Δ pairing cells and *RAD5^Tt^*Δ pairing cells (Figure 5). The persistence of γ-H2AX foci and the absence of H3K9ac, H3K18ac, and H3K56ac strongly suggests that both IswiG^Tt^ and Rad5^Tt^ play important roles in DNA repair involving euchromatin formation in the selected hMIC.

## 4. Discussion

### 4.1. Snf2 Family Proteins Localizing to the MAC

To identify the Snf2 family proteins involved in hMIC selection, we generated cells expressing fluorescent-tagged proteins for 10 Snf2 family proteins whose genes are actively transcribed in the early to mid-phase of conjugation and whose subcellular localization has not previously been investigated. Our observations revealed that Snf2 family proteins of *Tetrahymena*, with the exception of Rad54, exclusively localized to either the MAC or the MIC. In ciliates, active gene transcription occurs only in the MAC. This is responsible for maintaining cellular homeostasis. We found that IswiS^Tt^, Ino80^Tt^, Swr1^Tt^, Mot1^Tt^, and Rad16A^Tt^ localized exclusively to the MAC. According to previous studies in mammalian cells and yeast, Snf2 family proteins belonging to the Snf2 subfamily, Mi-2 subfamily, Chd7 subfamily, Iswi subfamily, Ino80 subfamily, Swr1 subfamily, and Mot1 subfamily are involved in transcription activation and repression [49,50,51,52,53]. In *Tetrahymena*, Brg1^Tt^ (Snf2 subfamily), Chd3^Tt^ (Mi-2 subfamily), and Chd7^Tt^ (Chd7 subfamily) exclusively appear in the MAC and play roles in transcription regulation [23,24]. It is reasonable to consider that IswiS^Tt^, Ino80^Tt^, Swr1^Tt^, and Mot1^Tt^, whose MAC localization was uncovered in this study, also contribute to transcription regulation. Fission yeast Rad16 forms a complex with Rad7 and functions in DNA repair in the genome-wide nucleotide-excision repair system [54]. Rad16A^Tt^ localizing to the MAC may participate in DNA repair and may contribute to the stability of the MAC genome.

### 4.2. Rad54

Rad54 plays an important role in the homologous recombination pathway, one of the major pathways of DNA repair [55]. Rad54 is required for genome stability, and mutations in Rad54 cause chromosome loss [56]. TTHERM_00237490 is a sole gene encoding Rad54 in the *Tetrahymena* genome, and its protein Rad54^Tt^ localized to both the MAC and MIC. We attempted to establish *RAD54^Tt^* knockout cells, but the *NEO4* cassette could not replace all somatic loci, indicating Rad54^Tt^ is indispensable for cell viability. We suggest that *Tetrahymena* Rad54 contributes to genome stability in both nuclei.

When the selected hMIC appeared, a strong fluorescence signal from Rad54^Tt^-EGFP was found in the nucleus (Figure 2 and Appendix A). Rad51^Tt^ localizes to the selected hMIC with the disappearance of γ-H2AX, indicating that a homologous recombination pathway contributes to repairing DNA lesions introduced by PM-DSB formation [5]. When Rad51-bound single-stranded DNA invades the complementary strand of donor DNA, Rad54 acts as a motor to relax the donor helix [57]. In the selected hMIC, Rad54^Tt^, together with Rad51^Tt^, may function in the repair of DNA lesions induced by PM-DSB formation.

### 4.3. Lodestar

Lodestar^Tt^ exclusively localized to the MIC in vegetative cells. During conjugation, Lodestar^Tt^ also localized to the nuclei generated from the MIC, such as hMICs, selected hMIC, pronuclei, and zygotic nuclei (Figure 2 and Appendix A). In *Drosophila*, *LODESTAR* mutations result in chromatin bridging at the anaphase of mitosis. Lodestar has therefore been considered to facilitate proper chromosome segregation during mitosis [58,59]. In *Tetrahymena*, the MAC divides by amitosis, whereas the MIC divides by mitosis [60]. During conjugation, the MIC undergoes multiple nuclear divisions, all of which are also mitotic [4]. Therefore, Lodestar^Tt^, which appeared exclusively in the MIC and nuclei generated from the MIC, may be functionally equivalent to *Drosophila* Lodestar and participate in the proper segregation of MIC chromosomes during mitotic nuclear divisions. Unlike the *Drosophila LODESTAR* mutant, however, *Tetrahymena LODESTAR^Tt^*Δ grew as well as the wild-type cells, and no abnormal phenotype was observed in conjugating knockout cells (Figure 3B,E), suggesting that the function of Lodestar^Tt^ is not essential for mitosis, or that there are other proteins that can compensate for the function of Lodestar^Tt^ in *Tetrahymena*.

### 4.4. Possible Functions of IswiG^Tt^ and Rad5^Tt^ for the hMIC Selection

In either *ISWIG^Tt^*Δ pairs or *RAD5^Tt^*Δ pairs, all hMICs were degraded, indicating that both IswiG^Tt^ and Rad5^Tt^ play important roles in the hMIC selection. Pair formation occurred in both gene knockout cells as in the wild-type cells, and their MIC began meiosis. No morphological abnormalities were found in the meiotic MIC chromosomes by orcein stain. These indicate that meiosis proceeded normally in both *ISWIG^Tt^*Δ and *RAD5^Tt^*Δ pairs. γ-H2AX foci were found in the hMICs of both knockout pairs, indicating that depletion of either IswiG^Tt^ or Rad5^Tt^ did not affect PM-DSB formation which is essential for the hMIC selection [5]. This indicated that the involvement of IswiG^Tt^ and Rad5^Tt^ in hMIC selection occurs after PM-DSB formation.

After PM-DSB formation, DNA repair factors exclusively localize to the selected hMIC in which γ-H2AX foci disappear [5]. At the same time, H3K9ac and H3K18ac, epigenetic markers for euchromatin formation, appear in the nucleus (Figure 5) [5]. By contrast, γ-H2AX foci persisted and neither H3K9ac nor H3K18ac appeared in all four hMICs in both the *ISWIG^Tt^*Δ and *RAD5^Tt^*Δ pairs (Figure 4 and Figure 5). Based on these results, we propose the following two hypotheses as the cause of the amicronucleate phenotype in the *ISWIG^Tt^*Δ pairing cells and *Rad5^Tt^*Δ pairing cells. First, depletion of either the IswiG^Tt^ or Rad5^Tt^ causes a failure of the hMIC selection, resulting in the degradation of all hMICs as unselected hMICs. Second is that hMIC selection itself occurs, but that DNA repair with euchromatin formation fails, resulting in the degradation of the selected hMIC as well as the unselected hMICs.

In the selected hMIC, Rad51/Rad54 and DNAPKcs appear simultaneously with the disappearance of γ-H2AX. This indicates that both non-homologous end-joining and homologous recombination repair pathways facilitate repair of DNA lesions caused by PM-DSBs. IswiG^Tt^ localized to the selected hMIC in which γ-H2AX disappeared (Appendix A). The mammalian Iswi containing complex is one of the major chromatin remodelers that function in three major DNA repair pathways: homologous recombination, non-homologous end-joining, and nucleotide excision repair [61]. Considering these, IswiG^Tt^ may contribute to DNA repair in the selected hMIC, which is carried out by Rad51/Rad54 and DNAPKcs rather than direct involvement in the hMIC selection. The amicronucleate phenotype observed in the *ISWIG^Tt^*Δ pairs may be due to defects in DNA repair and euchromatin formation in the selected hMIC.

Rad5^Tt^ localized to the selected hMIC in which γ-H2AX foci disappeared (Appendix A). This might suggest that Rad5^Tt^ also contributes to DNA repair in the selected hMIC. Indeed, yeast Rad5 and its human orthologs, HLTF and SHPRH, are required for tolerance to DNA damage [62]. However, the Rad5-dependent DNA repair pathway only works through DNA replication [63]. During the early-mid phase of *Tetrahymena* conjugation, DNA replication occurs at meiosis anaphase II or after the gametogenic mitosis, but DNA replication during the post-meiosis stage in which PM-DSB formation and its repair occurs has not been found [4]. If Rad5^Tt^ is involved in DNA repair and euchromatin formation in the selected hMIC, it may be an independent mechanism from DNA replication. A previous report indicates that Rad5 plays various biological functions, not limited to DNA repair through DNA replication [45]. The possibility that Rad5^Tt^ participates in DNA repair involving euchromatin formation via an unknown mechanism in the selected hMIC cannot be ruled out.

It should be noted that a part of *ISWIG^Tt^*Δ pairs and *RAD5^Tt^*Δ pairs could reach the progeny nuclear development stage (Figure 3E). This may result from incomplete replacement in KO strains, while it was not detected by PCR (Appendix A). Alternatively, this may indicate the existence of a pathway that can partially complement the loss of IswiG^Tt^ or Rad5^Tt^.

### 4.5. Tetrahymena Monitors the Appearance of the Selected hMIC in Which DNA Repair and Euchromatin Formation Are Completed

A recent study showed that Nrp1, which interacts with histone chaperone Asf1 and core histone proteins, localizes to the selected hMIC [17,18]. Nrp1 is required for DNA repair involving euchromatin formation in the selected hMIC. When Nrp1 is suppressed, γ-H2AX foci persist in all hMICs, and eventually, all hMICs are degraded [17]. Here, we showed that both IswiG^Tt^ and Rad5^Tt^ participate in DNA repair involving euchromatin formation in the selected hMIC. In *ISWIG^Tt^*Δ pairs and *RAD5^Tt^*Δ pairs, all hMICs are degraded. As a result, those knockout mutants showed an amicronucleate phenotype. Based on these observations, we suggest that there is a mechanism in the post-meiosis stage that monitors the emergence of mature pronuclei in which PM-DSBs are repaired, euchromatin formation is completed, and that protects mature pronuclei from nuclear degradation. When PM-DSB formation becomes aberrant, or localization of Nrp1, IswiG^Tt^, or Rad5^Tt^ to the selected nuclei fails, the DNA repair involving euchromatin formation in the selected nuclei that develop into pronuclei is not completed. Consequently, the selected hMIC degraded in the same manner as the unselected hMIC.

### 4.6. The Involvement of the Iswi Subfamily in Pronuclear Generation with Euchromatin Formation Is Evolutionarily Conserved

In some systems, the sperm paternal genome is packaged as a non-nucleosomal and highly compacted structure [64]. After the fertilizing sperm nucleus has entered the egg cytoplasm, the paternal genome is repackaged into a typical nucleosome structure by maternally providing canonical histone proteins [65]. During this process in *Drosophila*, maternal Iswi selectively accumulates into the male pronucleus and facilitates nucleosome repacking. Following maternal histone deposition, the chromatin structure in the male pronucleus becomes decondensed [66]. Here, we demonstrated that the Iswi subfamily protein was crucial for chromatin decondensation/euchromatin formation during gamete pronuclei generation in *Tetrahymena*. This implies that the functional association between Iswi subfamily proteins and gametogenesis is evolutionarily conserved.

## Figures and Tables

**Figure 1 microorganisms-10-02426-f001:**
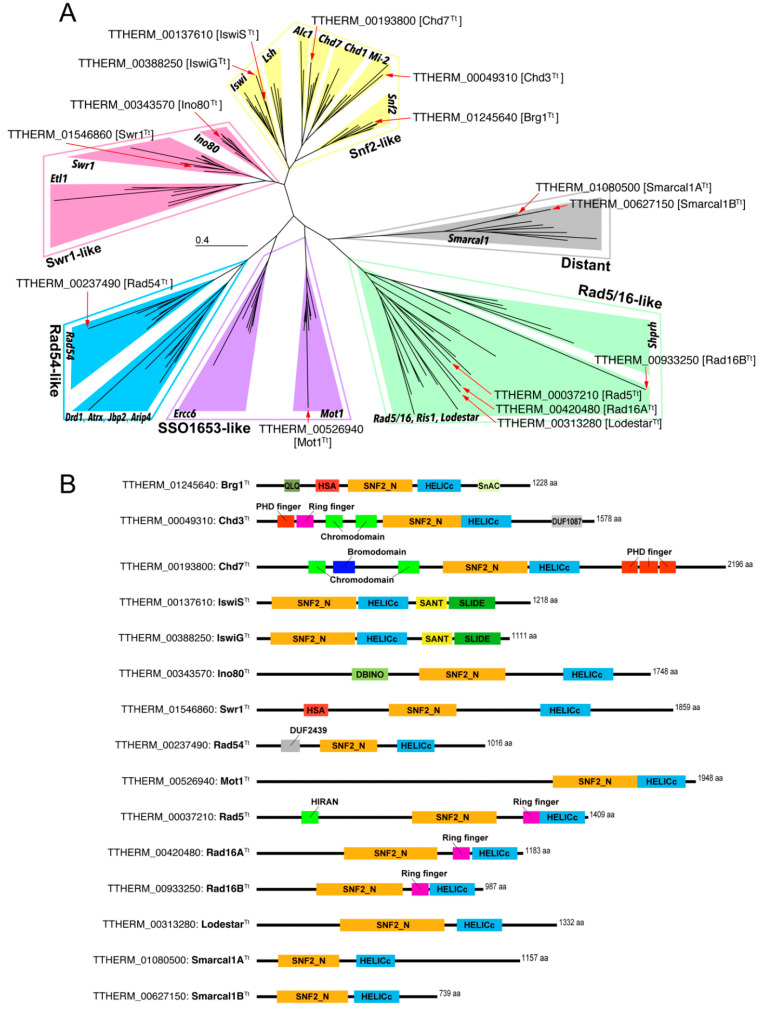
(**A**) An unrooted ML phylogenetic tree based on conserved amino acid sequences corresponding to the SNF_N and HELICc domains. Details including bootstrap values and proteins employed are displayed in Appendix A. Open boxes represent six general groupings. Bold oblique lines indicate subfamilies; (**B**) schematic diagram of the predicted domain/motif structure of 15 Snf2 family proteins found from *Tetrahymena*.

**Figure 2 microorganisms-10-02426-f002:**
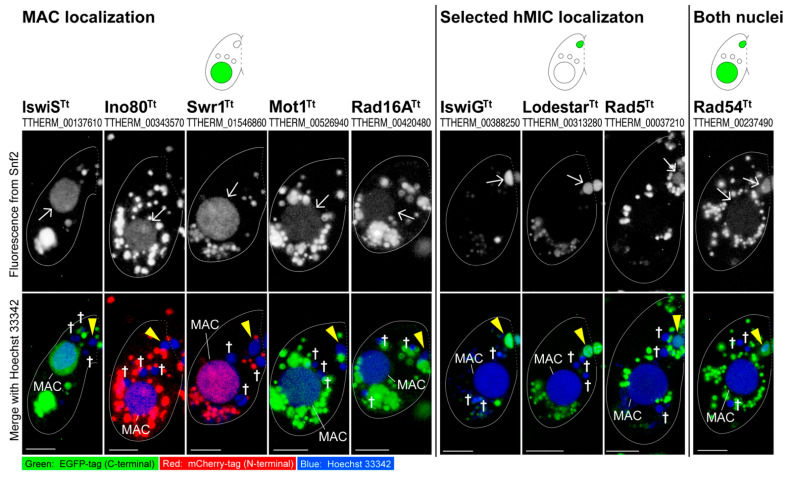
Subcellular localization of *Tetrahymena* Snf2 family fluorescent tagged proteins when the selected hMIC appeared. Arrow-fluorescent signal from EGFP/mCherry-tagged Snf2 family protein. The fluorescence signals scattered in the cytoplasm are likely the background derived from the fluorescent-tagged Snf2 family proteins undergoing degradation by turnover; triangle—the selected hMIC; †—unselected hMIC; the scale bar denotes 10 µm.

**Figure 3 microorganisms-10-02426-f003:**
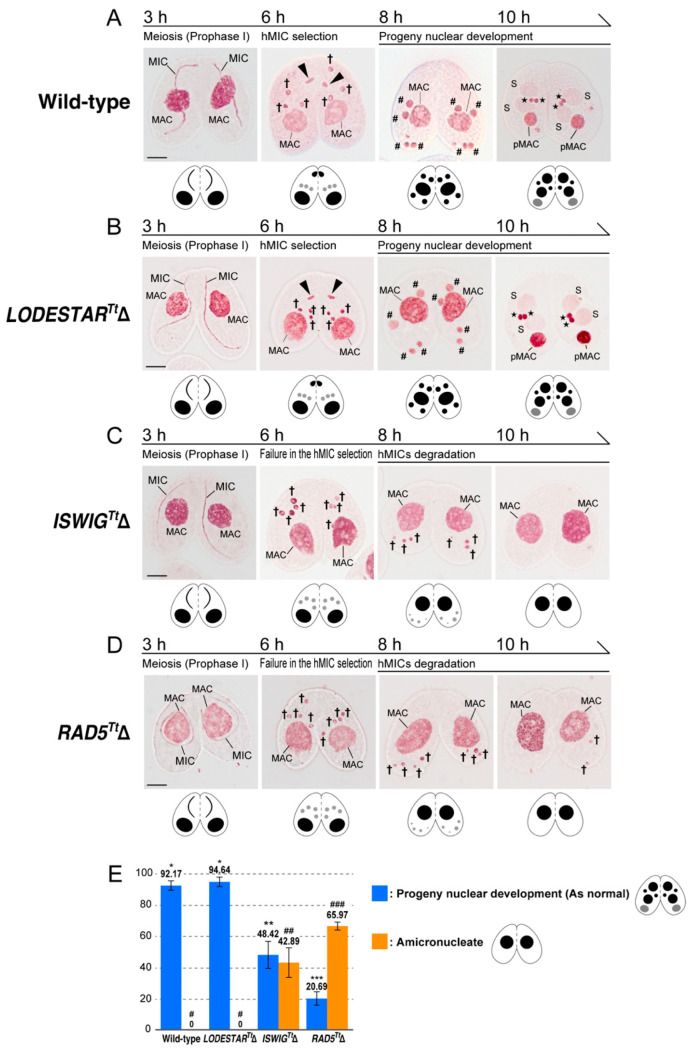
Knockout of the Snf2 family proteins which appear in the selected hMIC on conjugation. (**A**) acetic orcein staining of the wild-type pairs. Because acetic orcein stains DNA, nuclei are visualized under light microscopic observation; (**B**) acetic orcein staining of the *LODESTAR^Tt^*Δ pairs; (**C**) acetic orcein staining of the *ISWIG^Tt^*Δ pairs; (**D**) acetic orcein staining of the *RAD5^Tt^*Δ pairs. Schematic diagrams of nuclear behavior are shown below each micrograph. Triangle—the selected hMIC; †—unselected hMIC; #—zygotic nuclei undergoing post-zygotic nuclear divisions; S-progeny MAC Anlagen; star-progeny MIC; pMAC-degrading parental MAC. The scale bar denotes 10 µm; (**E**) percentage of pairs showing progeny nuclear development and amicronucleate phenotype at 10 h. Columns represent means. Difference in the number of symbols (asterisk-* and sharp-#) indicates a statistically significant difference (*p* > 0.01, One-way ANOVA with post-hoc Tukey HSD test).

**Figure 4 microorganisms-10-02426-f004:**
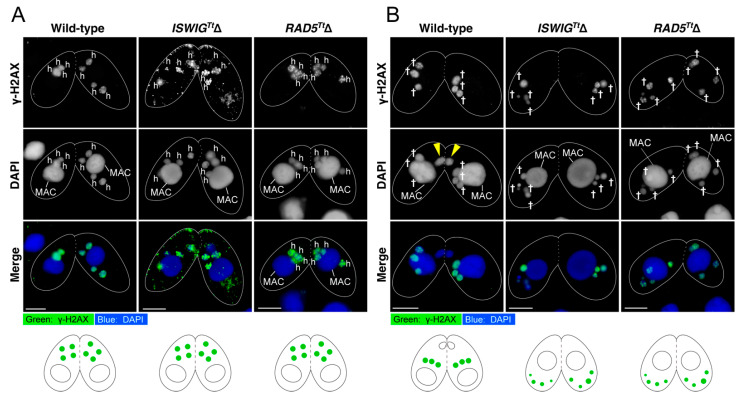
The post-meiotic γ-H2AX formation was not affected but persisted in the *ISWIG^Tt^*Δ pairs and the *RAD5^Tt^*Δ pairs. (**A**) γ-H2AX foci appeared at the post-meiosis stage; h-hMIC resulting from MIC meiosis; the scale bar denotes 10 µm. (**B**) γ-H2AX foci disappeared in the selected hMIC in wild type pairs at 6 h from the initiation of conjugation. In contrast, it persisted in degrading hMICs of the *ISWIG^Tt^*Δ pairs and the *RAD5^Tt^*Δ pairs in which hMIC selection failed. Schematic diagrams of the appearance of γ-H2AX foci are shown below each micrograph set. Triangle—the selected hMIC; †—unselected hMIC; the scale bar denotes 10 µm.

**Figure 5 microorganisms-10-02426-f005:**
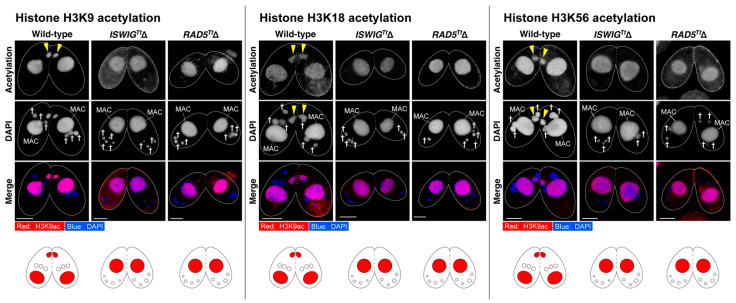
Signals for H3K9ac, H3K18ac, and H3K56ac appeared in the selected hMIC in which γ-H2AX dephosphorylation took place in the wild-type pair. In contrast, none of the hMIC underwent Histone H3 acetylation in the *ISWIG^Tt^*Δ pairs and the *RAD5^Tt^*Δ pairs in which hMIC selection failed. Schematic diagrams for acetylated Histone H3 are shown below each micrograph set. Triangle—the selected hMIC; †—unselected hMIC; The scale bar denotes 10 µm.

## Data Availability

Not applicable.

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
