# Peer review of "Snf2 Proteins Are Required to Generate Gamete Pronuclei in Tetrahymena thermophila"

_microorganisms, 2022, doi:10.3390/microorganisms10122426_

Round 1

Reviewer 1 Report

The authors perform a screen to determine which SNF2 family protein localizes specifically to the post-meiotic MIC that is retained post-meiosis in Tetrahymena (MIC selection).  Knockout analysis indicates two of the candidates are Important to the process of MIC selection.

This paper is well written and interesting.

Minor comments:

1.  With respect to transformation of tetrahymena –I am under the impression that transformation of Tt by electroporation was only possible by conjugating cells, by introducing the DNA ~8-10h post-mixing.  If indeed it is possible to electroporate non-conjugating cells, can the authors add the original reference where this was first demonstrated – If this is the first demonstration of this, could the authors mention this in the text and indicate what they have done differently to make this possible?

Line 271  -  could they refer to the figure where they show the localization as a signpost for reader ? Or at least indicate “see below” to note that this is the study that demonstrates this ?

Line 315/316 – “Each fusion protein with EGFP or mCherry was expressed from its endogenous MAC locus OR FROM the cadmium inducible MTT1 promoter” – see capitalized letters for suggested change. 

Figure 2 – the authors should acknowledge / address the high background fluorescence – is it vacuolar ?

Page 9 – for the non-specialist, could the authors indicate what the acetic orcein staining is being used for ?

Knockout analysis of the respective KOs – it is more conventional to use semi-quantitative real time POCR to assess whether there are any detectable transcripts of the knocked out gene – here the authors use PCR to assess whether they can detect any of the original WT locus – which is larger than the KO locus (PCR can be biased towards amplification of smaller fragments) - post-assortment. The authors should acknowledge that these strains may not be complete knockouts

The authors should cite PMID: 24120531 which was the first to demonstrate Asf1-Nrp1 physical interaction

Author Response

Could you please see the attachment?

Reviewer 2 Report

In this manuscript, Fukuda et al systematically studied the subcellular localization of Tetrahymena Snf2 family proteins that are expressed during the early to mid-phase of conjugation, by knocking in (KI) GFP- or mCherry-tagged gene copy. By co-staining with post-meiotic DSBs marks (represented by gH2A.X signals) and the euchromatin marks (representative by H3K9ac, H3K18ac, and H3K56ac signals), the authors revealed that four Snf2 family proteins (IswiGTt , Rad54Tt, Rad5Tt, and LodestarTt) appeared in the selected hMIC and that both IswiGTt and Rad5Tt were required for the hMIC selection followed by gametic pronuclei generation. The manuscript is well written and organized. The data is clear and convincing. This study provides important insights into how hMIC is selected in Tetrahymena.

A few points need to clear out before accept for publication.

1.      The gH2A.X signals, marking DSBs, initiate in Tetrahymena conjugation when the MICs start to elongate, before meiosis, which is different form the gH2A.X signal marking “post-meiosis DSBs”. The authors need to make this clear in the introduction and cite proper reference for the post-meiosis DSBs. The references citing this was not correct (page 1 line 46).

2.      The authors need to explain why most of the Snf2 genes were tagged with C-terminal GFP, while only two of them with N-terminal mCherry. In addition, it will be more convincing if the authors could at least provide one gene with both C-terminal GFP tag and N-terminal mCherry tag, showing that both KI versions of the tagged gene have the same subcellular localization.

3.      Can the authors explain why the post-meiotic γ-H2AX staining in IswiGTtΔ pairs showed scattered pattern in the cytoplasm in addition to the MIC?

Author Response

Could you please see the attachment?

Reviewer 3 Report

This is an exciting study addressing a very interesting event during Tetrahymena conjugation.  The study provides dramatic insight linking the cytological events:  'nuclear selection, preservation and gametogenic mitosis'  to the molecular events: DNA repair and euchromatin formation.  I have only one concern:    Figures 3A-D (and the accompanying text:  lines 389-391), suggest that NONE of the hMICs undergo gametogenic division, and they all migrate to the cell-posterior where they undergo autophagy.  Yet Figure 3E shows that 20-48% of the KO pairs DO exhibit normal nuclear behavior, even achieving conjugal endpoints.  This disconnect between data and text should be addressed.  As with many Tetrahymena phenotypes, we see incomplete penetrance for mysterious reasons.

Author Response

Could you please see the attachment?
